# Structural insights into the mycobacteria transcription initiation complex from analysis of X-ray crystal structures

Elizabeth A. Hubin[1,*], Mirjana Lilic[1,*], Seth A. Darst[1] & Elizabeth A. Campbell[1]

The mycobacteria RNA polymerase (RNAP) is a target for antimicrobials against tuberculosis, motivating structure/function studies. Here we report a 3.2 Å-resolution crystal structure of a *Mycobacterium smegmatis* (*Msm*) open promoter complex (RPo), along with structural analysis of the *Msm* RPo and a previously reported 2.76 Å-resolution crystal structure of an *Msm* transcription initiation complex with a promoter DNA fragment. We observe the interaction of the *Msm* RNAP α-subunit C-terminal domain (αCTD) with DNA, and we provide evidence that the αCTD may play a role in *Mtb* transcription regulation. Our results reveal the structure of an Actinobacteria-unique insert of the RNAP β' subunit. Finally, our analysis reveals the disposition of the N-terminal segment of *Msm* σ^A, which may comprise an intrinsically disordered protein domain unique to mycobacteria. The clade-specific features of the mycobacteria RNAP provide clues to the profound instability of mycobacteria RPo compared with *E. coli*.

[1] The Rockefeller University, 1230 York Avenue, New York, New York 10065, USA. * These author contributed equally to this work. Correspondence and requests for materials should be addressed to S.A.D. (email: darst@rockefeller.edu) or to E.A.C. (email: elizabeth.campbell0@gmail.com).

The infectious disease tuberculosis (TB), caused by *Mycobacterium tuberculosis* (*Mtb*), claims almost 2 million lives annually. Efforts to combat TB are impeded by the increase of multi-drug resistant *Mtb* strains. Rifamycins, an important component of modern TB therapy[1], target the *Mtb* RNA polymerase (RNAP), the enzyme responsible for all transcription in the bacterium. This provides an incentive to determine structures of the mycobacteria RNAP to aid in the development of improved therapeutics.

In bacteria, transcription initiation occurs when the $\sim 400$ kDa RNAP catalytic core enzyme (E, subunit composition $\alpha_2\beta\beta'\omega$) associates with the promoter specificity subunit, $\sigma^A$, to create the holoenzyme ($E\sigma^A$), which directs the enzyme to promoter DNA sites through sequence-specific recognition of the $-35$ and $-10$ promoter elements by $\sigma^A$ domains 4 ($\sigma^A_4$) and 2 ($\sigma^A_2$), respectively[2]. Initial $E\sigma^A$/promoter DNA recognition triggers a series of events as the enzyme unwinds 12–14 bp of DNA to form the transcriptionally competent open promoter complex (RPo)[3,4]. This functional paradigm was developed through studies of the RNAP from *Escherichia coli* (*Eco*)[3]. While structures of *Eco* RNAP are available[5–7], most high-resolution structures of bacterial RNAPs in different states of the transcription cycle come from *Thermus* RNAPs[8]. Recent mechanistic studies have revealed that mycobacteria RNAP exhibits unexpected differences in basic and regulated functions from those of *Eco* and *Thermus*. These include differences in: (1) termination signals and the influence of elongation factors[9]; (2) the kinetic landscape during initiation[4,10–12]; (3) the dependence on essential general transcription factors such as CarD (absent in *Eco*) and RbpA (absent in both *Eco* and thermus)[13,14] and (4) the structures and insertion points of lineage-specific inserts in the RNAP large subunits[15].

Previously we described a 2.76 Å-resolution crystal structure of *M. smegmatis* (*Msm*) $E\sigma^A$ with RbpA and bound to an upstream fork (us-fork) promoter DNA fragment (RbpA/$E\sigma^A$/us-fork), focusing on the roles of the essential transcription factors RbpA and CarD in the initiation process[4]. Here, we present a 3.2 Å-resolution crystal structure of a full *Msm* RPo containing RbpA/$E\sigma^A$ and promoter DNA containing a complete transcription bubble and a 4-mer RNA hybridized to the DNA template strand (t-strand) in the RNAP active site. Analysis of these two structures, focusing on RNAP structural features not addressed in our previous manuscript[4], provides several highlights, including: (1) the highest resolution view available of conserved RNAP/DNA interactions in RPo, (2) the interaction of the *Msm* RNAP $\alpha$-subunit C-terminal domain ($\alpha$CTD) with an AT-rich segment of the DNA along with evidence that the $\alpha$CTD may play a previously unappreciated role in mycobacteria transcription regulation, (3) the structure of lineage-specific insert $\beta'$i1, unique to Actinobacteria[15] and (4) the disposition and unique features of the N-terminal segment of *Msm* $\sigma^A$, unique to mycobacteria (termed $\sigma^A_N$ here).

## Results

**Overall structure of *Msm* transcription initiation complexes.** Structure determination of the *Msm* RbpA/$E\sigma^A$ with an us-fork promoter fragment (Supplementary Fig. 1) was previously described[4]. We formed a complete RPo by combining *Msm* RbpA/$E\sigma^A$ with a duplex promoter DNA scaffold ($-37$ to $+13$ with respect to the transcription start site at $+1$) but with a non-complementary transcription bubble generated by altering the sequence of the t-strand DNA from $-11$ to $+2$ plus an RNA primer complementary to the t-strand DNA from $+1$ to $-3$, yielding a 4 bp RNA/DNA hybrid (Fig. 1a). The *Msm* RbpA/RPo crystallized in the same space group (P2$_1$) with very similar unit

cell parameters as the us-fork complex and diffraction data were collected to a resolution of 3.2 Å. The RPo structure was solved by molecular replacement and refined (Fig. 1b; Supplementary Table 1). There were no significant conformational differences between the two structures, which superimposed with a root-mean-square deviation (r.m.s.d.) of 0.59 Å over 2,933 $\alpha$-carbons.

**Protein–DNA interactions.** Interactions of RNAP with the full transcription bubble and the upstream double-strand/single-strand (ds/ss) DNA junction at the upstream edge of the $-10$ element where transcription bubble formation initiates in RPo have only been visualized at 4 Å resolution[16]. The structures reported here at much higher resolution confirm and extend these previous observations (Figs 1c and 2a, Supplementary Figs 2–4).

As seen previously[16], the invariant W-dyad of $\sigma^A_2$ (*Msm* $\sigma^A$ W287/W288; Supplementary Fig. 2D) maintains the ds/ss ($-12/-11$) junction at the upstream edge of the transcription bubble (Fig. 1c). The W-dyad forms a 'chair'-like structure, with W287 serving as the back of the chair, and W288 as the seat, buttressing the T$_{-12}$(nt) from the major groove side. The methyl group of the T$_{-12}$(nt) base approaches the face of the W288 side chain at a nearly orthogonal angle, forming a favourable methyl-$\pi$ interaction[17,18] (Fig. 1c).

Arg residues of $\sigma^A_2$ support the role of the W-dyad in stabilizing the upstream ds/ss junction by buttressing the Trp side chains from the face opposite the T$_{-12}$(nt) base (Fig. 1c). The guanidino group of R290 (absolutely conserved among Group 1 $\sigma$'s; Supplementary Fig. 2D) forms a cation-$\pi$ interaction[19] with the downstream face of W287, sandwiching the W287 indole side chain between the exposed T$_{-12}$(nt) base with an inter-ring angle of $\sim 27°$ (Fig. 1c, Supplementary Fig. 4). R268 reaches across from the $\sigma^A_2$ region 2.2 $\alpha$-helix to make salt bridges with the $-13$ nt and $-14$ nt phosphates, positioning the R268 hydrophobic alkyl chain to interact with the underside of the W288 side chain (opposite the methyl-$\pi$ interaction with the T$_{-12}$(nt) base; Fig. 1c). The position corresponding to *Msm* $\sigma^A$ R268 is conserved as either K or R (Supplementary Fig. 2D). In the 4 Å-resolution *Taq* RPo structure (PDB ID 4XLN (ref. 16)), the electron density for both of the corresponding Arg side chains (*Taq* $\sigma^A$ R237/R259) was weak and interactions with the W-dyad were not observed. The ionic strength of the *Taq* RPo crystallization solution was $\sim 4.9$ M [1.6 M (NH$_4$)$_2$SO$_4$], while the ionic strength of the *Msm* RbpA/$E\sigma^A$ transcription initiation complexes (TICs) was $\sim 0.6$ M (polyethylene glycol $+0.2$ M Li$_2$SO$_4$). The electrostatic interactions formed by these two Arg residues (corresponding to *Msm* $\sigma^A$ R268/R290; *Taq* $\sigma^A$ R237/R259) were likely weakened in the high ionic strength of the *Taq* RPo crystallization solution.

Additional RNAP/promoter DNA interactions were described at 4 Å resolution but are now observed much more clearly (Supplementary Figs 2C, 3,4). A summary of the RNAP/promoter DNA interactions is shown schematically in Fig. 2b.

**A mycobacteria $\alpha$CTD–DNA interaction.** During refinement of the *Msm* RbpA/$E\sigma^A$/us-fork complex[4], unaccounted difference density appeared that corresponded to an $\alpha$CTD bound to an A/T-rich region of the promoter DNA from $-29$ to $-24$ (A$_{-29}$AAGTG$_{-24}$; Fig. 3). The $\alpha$CTD structure was built and refined previously but was not addressed in the earlier manuscript[4]. Very weak and broken electron density for the $\alpha$CTD was also observed in the *Msm* RPo structure, indicating very low occupancy. The $\alpha$-N-terminal domain (essential for $\alpha$ dimerization and RNAP assembly)[20,21] is flexibly linked to the $\alpha$CTD by a $\sim 25$ residue unstructured linker. In *Eco*, the $\alpha$CTD binds to A/T-rich regions upstream of the promoter $-35$

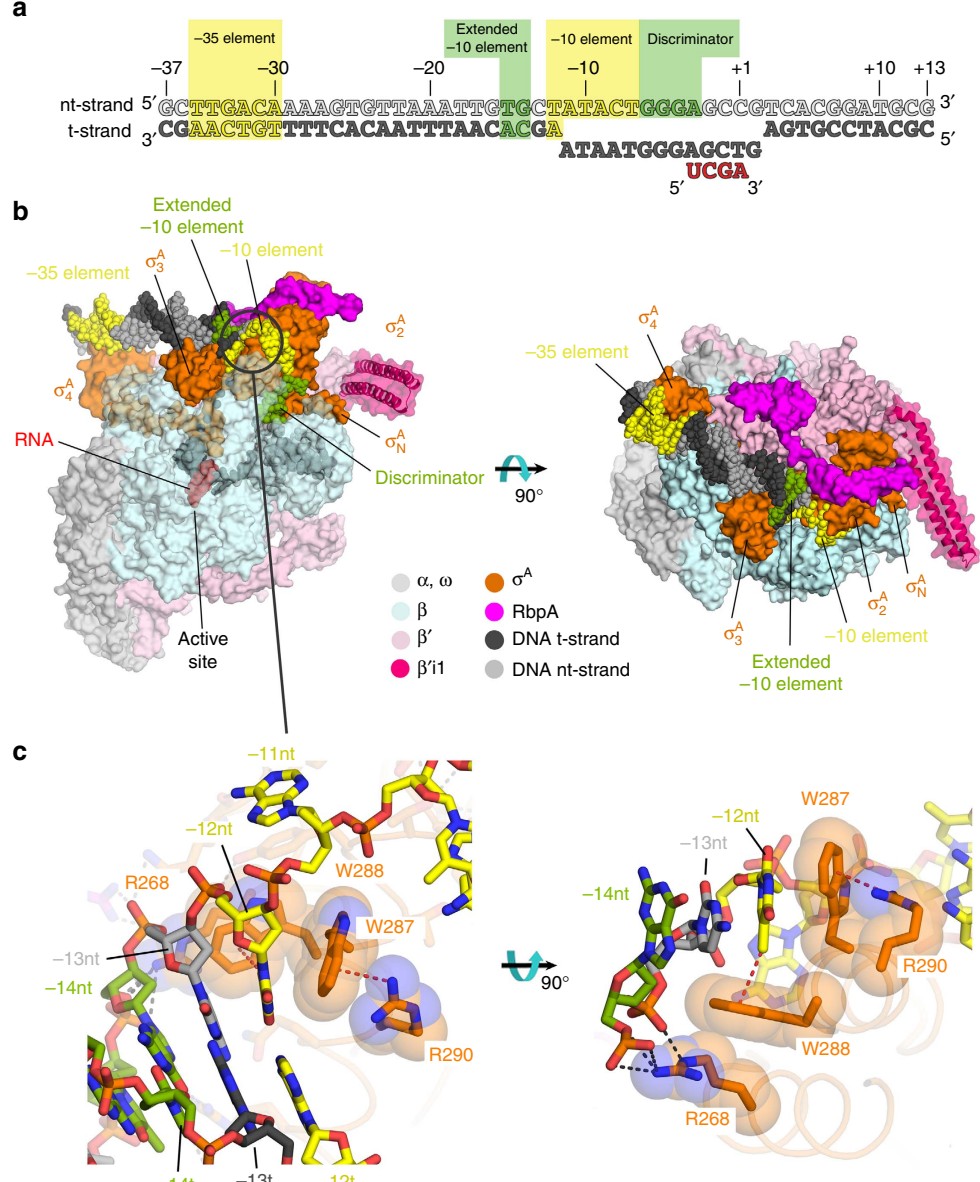

**Figure 1 | Structure of the *Msm* RbpA/RPo.** (**a**) Synthetic oligonucleotides used for the *Msm* RbpA/RPo crystallization. The numbers above denote the DNA position with respect to the RNA transcript 3′-end (+1). The DNA sequence is derived from the full con promoter[63]. The nt-strand DNA (top strand) is coloured light grey; the t-strand DNA (bottom strand), dark grey; RNA, red. The −35 and −10 elements are shaded yellow. The extended −10 (ref. 38) and discriminator[64,65] elements are coloured green. (**b**) Overall structure of the *Msm* RbpA/RPo. The color-coding of most of the structural features is denoted in the legend. Protein components (core RNAP, σ$^A$, RbpA) are shown as molecular surfaces. The surface of the RNAP β subunit is transparent, revealing the nucleic acids and σ$^A$ elements located in the RNAP active centre cleft. The surface of the lineage-specific insert β′i1 is transparent, revealing the α-carbon backbone ribbon underneath. The nucleic acids are shown as CPK atoms, coloured as in Fig. 1a. The circled region in the left view is magnified in Fig. 1c. (**c**) Magnified views showing the upstream ds/ss junction of the transcription bubble in RPo (obscuring elements of the structure have been removed). Side chains of the absolutely conserved σ$^A$ W-dyad (W288/W287) and conserved σ$^A$ Arg residues that buttress the W-dyad (R268, R290) are shown in orange along with transparent CPK spheres (Supplementary Fig. 2D). Polar interactions (hydrogen bonds, salt bridges) are shown as grey dashed lines. The cation-π interaction[19] (R290-W287) and the methyl-π interaction[17,18] [T$_{−12}$(nt)-W288] are shown as red dashed lines.

element (UP elements), activating transcription at many promoters[22]. The *Eco* αCTD also plays a major role in interacting with transcription factors[23,24]. The role, or even existence, of UP elements in mycobacteria transcription regulation has not, to our knowledge, been identified.

The location of the *Msm* αCTD and its relationship to neighbouring molecules in the crystal packing environment indicates that the αCTD belongs to a symmetry-related RNAP (Supplementary Fig. 5A) and is bound to the DNA adventitiously

(αCTD$_{symm}$) in a non-physiologically relevant position of the promoter between the −10 and −35 elements (Fig. 3a,b) rather than the physiologically relevant position upstream of the −35 element[22]. The structure of the *Msm* αCTD is essentially identical to the *Eco* αCTD (r.m.s.d. of 0.586 Å over 40 Cα's)[25]. All of the αCTD DNA-interacting residues are conserved between *Msm*, *Mtb* and *Eco*, and the αCTD/DNA interactions are essentially identical, including an ordered water molecule that mediates interactions between *Msm*/*Eco* R259/R265, N288/N294 and the

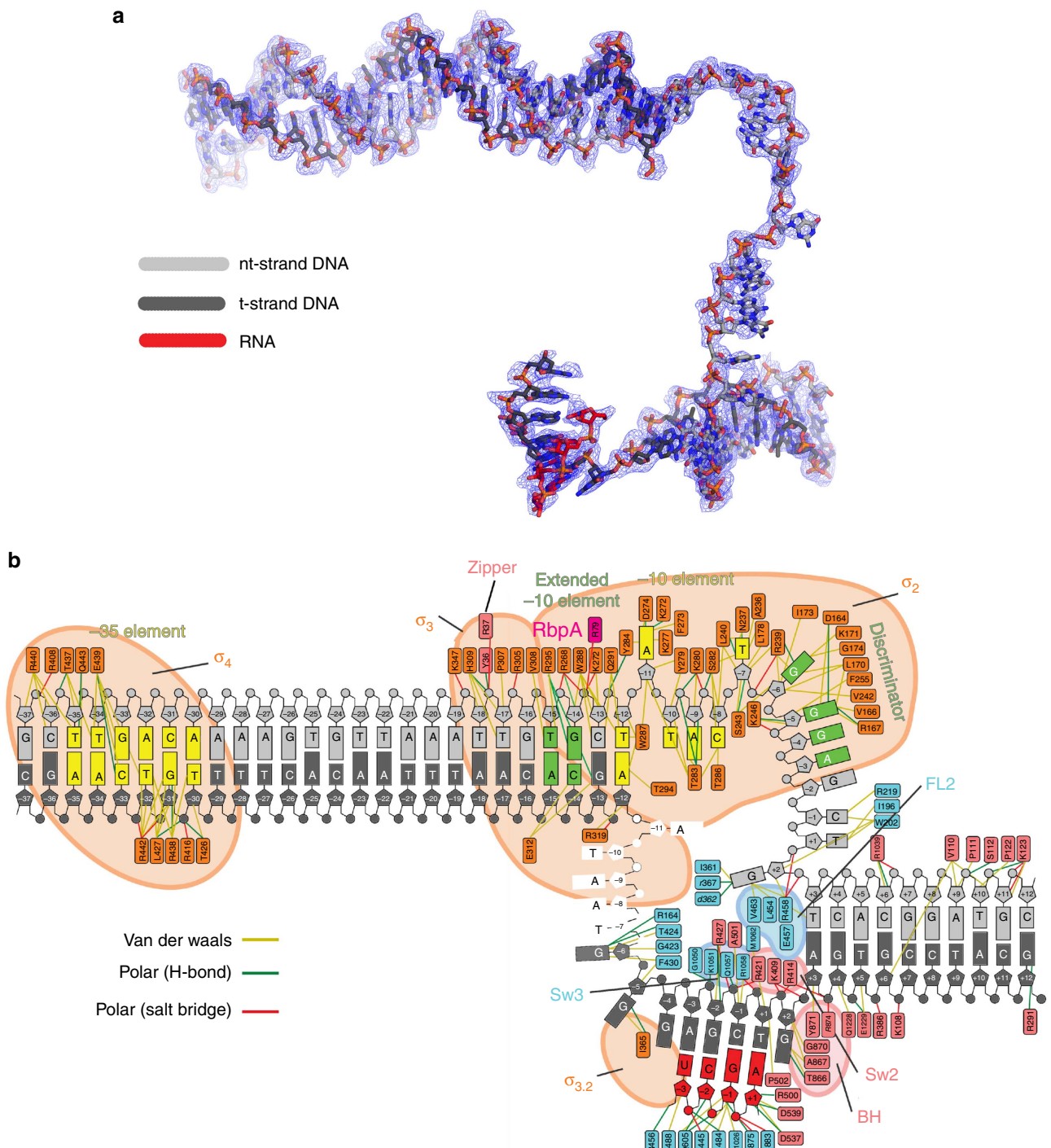

**Figure 2 | *Msm* RPo electron density and protein–DNA interactions.** (**a**) Electron density and model for RPo nucleic acids. Blue mesh, 3.2 Å resolution $2F_o - F_c$ map for nucleic acids (contoured at 1.0σ). (**b**) The nucleic acids shown are from the RPo structure (Fig. 1a). The protein–DNA interactions for the −10 element and upstream were derived from the 2.76 Å-resolution *Msm* RbpA/Eσ$^A$/us-fork complex (PDB ID 5VI8)[4], but only a small handful of these interactions differed in the 3.2 Å-resolution RPo structure. The protein/nucleic acid interactions for nucleic acids downstream of the −10 element were derived from the 3.2 Å-resolution RPo structure (Figs 1b and 2a). The t-strand DNA from −7 to −11 was disordered and not modelled. Protein/DNA interactions were defined as follows: van der Waals (≤ 4.5 Å), yellow lines; H-bonds (≤ 3.5 Å); salt bridges (≤ 4.5 Å), red lines.

DNA (Fig. 3c–e). Finally, *Eco* αCTD D259 and E261 have been shown to facilitate UP-element function by interacting with σ$^{70}$ R603 (ref. 26); all of these residues are conserved in the *Msm* and *Mtb* proteins (αCTD D253/D255, Fig. 2e, and σ$^A$ R457) but do not interact due to the non-physiological position of the αCTD.

**UP-element-like sequences are enriched in *Mtb* promoters**. The striking similarities between *Eco* and *Msm* αCTD structure and DNA interactions (Fig. 3) raises the possibility that αCTD/UP-element interactions play a role in mycobacteria transcription regulation. To further explore this idea, we used RNA-seq with transcriptional start site (TSS) mapping data to compare

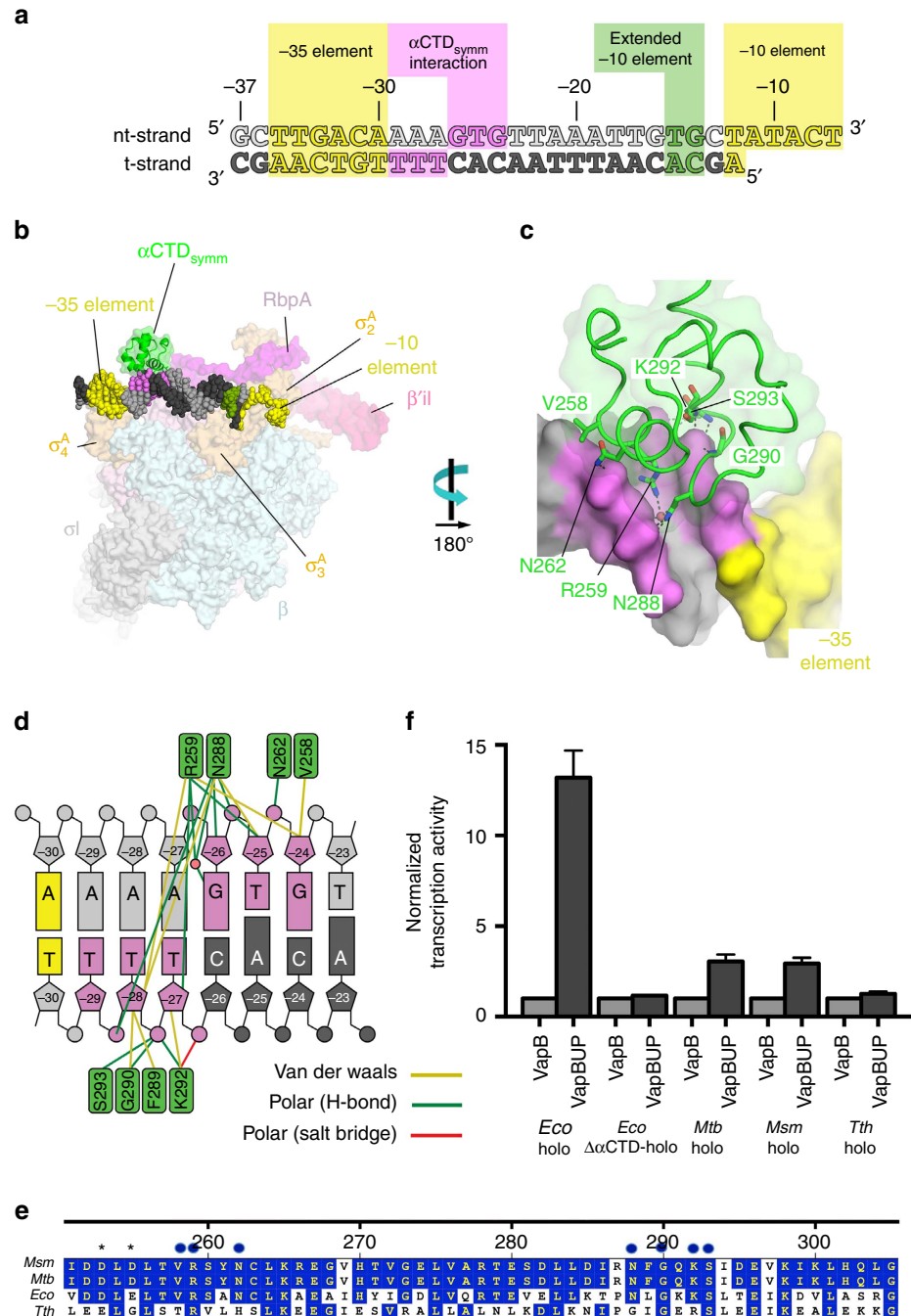

**Figure 3 | Mycobacteria αCTD/DNA interactions.** (**a**) Sequence of the us-fork promoter fragment used in the 2.76 Å-resolution *Msm* TIC structure[4]. The numbers above denote the DNA position with respect to the transcription start site ( +1). The DNA sequence is derived from the full con promoter[63]. The nt-strand DNA (top strand) is coloured light grey; the t-strand DNA (bottom strand), dark grey. The −35 and −10 elements are shaded yellow. The extended −10 (ref. 38) is coloured green. The DNA nts interacting with a symmetry-related αCTD (αCTD_symm) are coloured violet. (**b**) Overall view of the *Msm* RbpA/Eσ^A/us-fork complex. The color-coding is the same as Fig. 1b. The RNAP is shown as a molecular surface. The DNA is shown in CPK format, color-coded as Fig. 2a. Also shown is αCTD_symm (green backbone ribbon with transparent green molecular surface). (**c**) Close-up of αCTD_symm/DNA interactions (viewed from the back side of Fig. 2b). The αCTD_symm is shown as a green backbone worm with conserved DNA-interacting side chains shown (polar interactions are shown as grey dashed lines). A well-ordered water molecule mediating αCTD_symm interactions is shown as a pink sphere. The DNA is shown as a molecular surface. (**d**) Schematic illustrating αCTD_symm/DNA interactions. (**e**) Sequence alignment of *Msm*, *Mtb*, *Eco* and *Tth* αCTDs. The numbering at the top denotes the *Msm* and *Mtb* numbering. Residues conserved in at least three of the seqences are shaded blue. Residues that interact with the DNA are denoted with blue dots above the sequences. The DNA-interacting residues are all conserved between *Msm*, *Mtb*, and *Eco*, but not with *Tth*. Negatively charged residues shown to be important for *Eco* αCTD/σ^70_4 interactions that play a role in stimulating transcription are are denoted with asterisks (*) above. We infer that these residues also play a role in stimulatory *Msm* and *Mtb* αCTD/σ^A_4 interactions as well since they are conserved as negatively charged residues in *Msm* and *Mtb*. (**f**) Histogram showing abortive transcription activity of various holoenzymes (shown below) on the *Mtb* VapB promoter (normalized to 1) and VapBUP (engineered to contain a proximal UP-element sequence (Supplementary Fig. 6B,C). The values are the average of triplicate experiments. The error bars denote the s.e.m.

**Table 1 | Promoter motifs in *Eco* and *Mtb*.**

| Element | Sequence motif | *Eco* (3,746) | | | *Mtb* (1,775) | | |
|---|---|---|---|---|---|---|---|
| | | Hits | Expected random hits | Enrichment over random | Hits | Expected random hits | Enrichment over random |
| −10 element | ANNNT(6 − 9 N) | 2,953 | 680 | 4.3 | 1,702 | 53 | 32 |
| Ext. −10 element | TGNNANNNT | 530 (18%) | 185 (6.3%) | 2.9 | 256 (15%) | 96 (5.6%) | 2.7 |
| −35 element | TKKHNN(16 N)NANNNT | 346 (12%) | 137 (4.6%) | 2.5 | 69 (4.1%) | 47 (2.8%) | 1.5 |
| | TKKHNN (17 N)NANNNT | 583 (20%) | 137 (4.6%) | 4.3 | 146 (8.6%) | 47 (2.8%) | 3.1 |
| | TKKHNN (18 N)NANNNT | 330 (11%) | 137 (4.6%) | 2.4 | 116 (6.8%) | 47 (2.8%) | 2.5 |
| | All −35 elements | 1,259 (43%) | 411 (14%) | 3.1 | 331 (19%) | 141 (8.3%) | 2.3 |
| | −35 and ext. −10 elements | 172 (5.8%) | 26 (0.88%) | 6.6 | 44 (2.6%) | 8 (0.47%) | 5.5 |
| UP-element | AAANNNNNNNNNN(16 N)NANNNT | 145 (4.9%) | 44 (1.5%) | 3.3 | 21 (1.2%) | 8.7 (0.51%) | 2.4 |
| | AAANNNNNNNNNN(17 N)NANNNT | 150 (5.1%) | 44 (1.5%) | 3.4 | 17 (1.0%) | 8.7 (0.51%) | 2.0 |
| | AAANNNNNNNNNN(18 N)NANNNT | 144 (4.9%) | 44 (1.5%) | 3.3 | 16 (0.94%) | 8.7 (0.51) | 1.8 |
| | All UP elements | 439 (15%) | 132 (4.5%) | 3.3 | 54 (3.2%) | 26 (1.5%) | 2.1 |

N: any nt. K: G/T. H: not G.

the occurrence of promoter motifs upstream of TSSs in the *Eco* (ref. 27) and *Mtb* (ref. 28) genomes. We searched DNA sequences[29] within 50 bp upstream of the identified TSSs for motifs (denoted in Table 1), taking into account the variability in spacing between the −10 element and the TSS[30] and between the −10 and −35 elements[31]. We note that the search parameters (Table 1) are fairly restrictive and are expected to miss divergent promoters, so our analysis yields lower bounds on the estimates of promoter motif occurrence. Nevertheless, these searches provide a good basis for direct comparison of global promoter architecture in the two genomes. We note that the *Eco* and *Mtb* genomes have very different GC contents [*Eco*, 50.8% GC[32]; *Mtb*, 65.6% (ref. 33)] so promoter elements such as the A/T-rich UP-element[34] may have diverged between *Eco* and *Mtb*.

(1) The −10 element[31,35,36]: Most of the conservation in the −10 element is captured by the ANNNT motif[31,37], and the spacing to the TSS varies between 6 and 9 nt (ref. 30). Thus, we searched upstream of the 3,746 *Eco* and the 1,775 *Mtb* TSSs for an ANNNT (6–9 N) motif and identified 2,953 (*Eco*) and 1,702 (*Mtb*) −10 elements (Table 1). The −10 element is the most highly conserved[31] and the only essential element for Group 1 σ factor promoters, so the searches for the remaining promoter elements were done in the context of the −10 element hits for each genome.

(2) Extended −10 element[38]: The *Eco* and *Mtb* genomes utilize the extended −10 motif (TGNNANNNT) to very similar extents. In *Eco*, 18% of the 2,953 −10 element hits used an extended −10 element, while in *Mtb*, 15% of the 1,702 −10 element hits used an extended −10 element (Table 1). In both genomes, the extended −10 motif was found at nearly three times the expected frequency for chance occurrence of the motif.

(3) −35 element[31]: Previous analyses of the *Eco* and *Mtb* TSS data did not take into account the known variability in spacing between the −10 and −35 elements. The optimal −10/−35 spacing is 17 nt, and spacings of 16, 17 and 18 nt account for more than 75% promoters analysed by Shultzaberger *et al.*[31]. We searched for a −35 element motif comprising T(G/T)(G/T)(A/C/T)NN (which accounts

for most of the sequence conservation of the motif)[31] spaced 16, 17 or 18 nt upstream of the −10 element (Table 1). We found that *Eco* makes more extensive use of the paridigmatic −10/−35 promoter architecture, with 43% of the −10 element hits harbouring the −35 element compared to 19% in *Mtb* (Table 1).

(4) UP element[34]: UP-element sequences are highly divergent, essentially comprising A/T-rich sequences upstream of the −35 element[22]. However, Estrem *et al*[34]. used a SELEX approach to identify UP-element consensus sequences. We probed for the possible occurrence of UP element regulatory sequences in *Mtb* compared to *Eco* by searching for the most prominent feature of the proximal UP element, an 'AAA' motif 26, 27 or 28 nt upstream of the −10 element[34]. According to this stringent criterium, 15% of the *Eco* −10 element promoters harboured an UP element, while 3.2% of the *Mtb* promoters did (Table 1). While the occurrence of the UP element motif in *Mtb* appears to be small, the high GC content of the *Mtb* genome makes the 'AAA' motif highly unlikely to occur by chance; the 15% UP-element occurrence in *Eco* is 3.3-fold enriched over random, while the 3.2% UP-element occurrence in *Mtb* is 2.1-fold enriched over random (Table 1).

**An UP-element-like sequence enhances *Mtb* transcription.** Our promoter motif searches indicate that UP element sequences are enriched upstream of *Mtb* promoters. To test if a consensus proximal UP element can play a role in activating mycobacteria transcription, we engineered a native *Mtb* promoter, the vapB10p antitoxin promoter (VapB)[28] to contain a proximal UP-element sequence (VapBUP; Supplementary Fig. 5B). We compared transcription activity of VapB with VapBUP using *Eco* RNAP (positive control), *Eco* ΔαCTD-RNAP (negative control), and *Mtb* and *Msm* RNAPs. Transcription by *Eco* Eσ[70] was stimulated more than tenfold by the presence of the UP element, while *Eco* ΔαCTD-Eσ[70] was not (Fig. 3f), consistent with the role of the αCTDs in UP element activation[22]. Both *Mtb* and *Msm* holoenzymes were stimulated roughly threefold by the presence of the UP element (Fig. 3f), suggesting that αCTD/UP element interactions play a role in regulating mycobacteria transcription. Finally, we tested *Tth* RNAP on the same pair of promoters. Some

of the DNA-binding residues of the αCTD are not conserved in *Tth* αCTD (Fig. 2e), and the presence of the UP-element did not stimulate *Tth* RNAP transcription (Fig. 3f).

**An Actinobacteria-specific insertion in the β′ subunit.** The RNAP β and β′ subunits comprise linearly arranged segments of sequence conserved between all five kingdoms of life[15]. These conserved segments are separated by spacer regions that are not conserved between phylogenetically distinct groups of bacteria and large, lineage-specific domain insertions can occur within the spacer regions[15,39]. The Actinobacteria RNAPs, which includes *Msm* and *Mtb* RNAPs, contain one lineage-specific insertion, a ~90 residue insertion at about position 140 of the β′ subunit (β′i1)[15]. The insertion point of the Actinobacteria β′i1 is identical to the insertion point of an unrelated lineage-specific insertion, β′i2 of deinococcus-thermus[15] (Supplementary Fig. 6). The deinococcus-thermus β′i2 comprises five sandwich-barrel hybrid motifs[39] with complex topological connections and plays a role in σ[A] binding[40] (Supplementary Fig. 6A).

The *Msm* TIC structures show that β′i1 spans roughly residues 140–230 and folds into two long anti-parallel α-helices (β′i1-α1, residues 141–186; β′i1-α2, residues 191–228; Figs 1b and 4a,b). The two helices emerge from the tip of the RNAP clamp module and extend across the entrance to the RNAP active centre cleft (Figs 1b and 4a,b). An analysis of the electrostatic surface charge distribution revealed an asymmetry, with a positively charged surface facing the RNAP active site cleft and a negatively charged surface facing

outwards (Fig. 4b). A Blast search against *Msm* β′ residues 130–240 identified 720 homologues, all from Actinobacteria. Sequence alignments revealed a pattern of conserved charged residues: the first β′i1 α-helix (β′i1-α1, distal to the entrance to the RNAP active site cleft; Fig. 4b,c) contains a net charge of −8, while the β′i1-α2 helix (facing the entrance to the RNAP active site cleft) contains a net charge of +2 (Fig. 4c). Thus, the asymmetric charge distribution is a conserved structural feature of the Actinobacteria β′i1, suggesting a functional role.

**Group 1 σ N-terminal extension.** Group 1 σ's comprise three conserved structured domains (σ$_2$,σ$_3$ and σ$_4$)[41] and one divergent N-terminal extension (σ$^A_N$). Group 1 σ$^A_N$'s vary greatly in length (residues N-terminal of conserved region 1.2: *Mtb* σ$^A_N$, 225; *Msm* σ$^A_N$, 163; *Eco* σ$^{70}_N$, 95) and are not conserved across all clades (see below).

The *Eco* Group 1 σ, σ$^{70}$, is autoregulated by σ$^{70}_{1.1}$, which serves to prevent σ$^{70}$ interactions with promoter DNA in the absence of RNAP[42–44] and also plays a role in the formation of RPo[45,46]. Based on solution FRET and structural studies, *Eco* σ$^{70}_{1.1}$ is located within the RNAP active-site channel in Eσ$^{70}$, but in RPo, σ$^{70}_{1.1}$ is displaced outside the channel by the entering promoter DNA[6,47]. *Eco* σ$^{70}_{1.1}$ comprises a compact three-helical domain linked to the rest of σ$^{70}$ by a 37-residue linker, facilitating the large movements of σ$^{70}_{1.1}$ during RPo formation[6,44]. The sequence of *Eco* σ$^{70}_{1.1}$ is conserved among many bacterial Group 1 σ's (Supplementary Fig. 7A,B) but not universally so: the

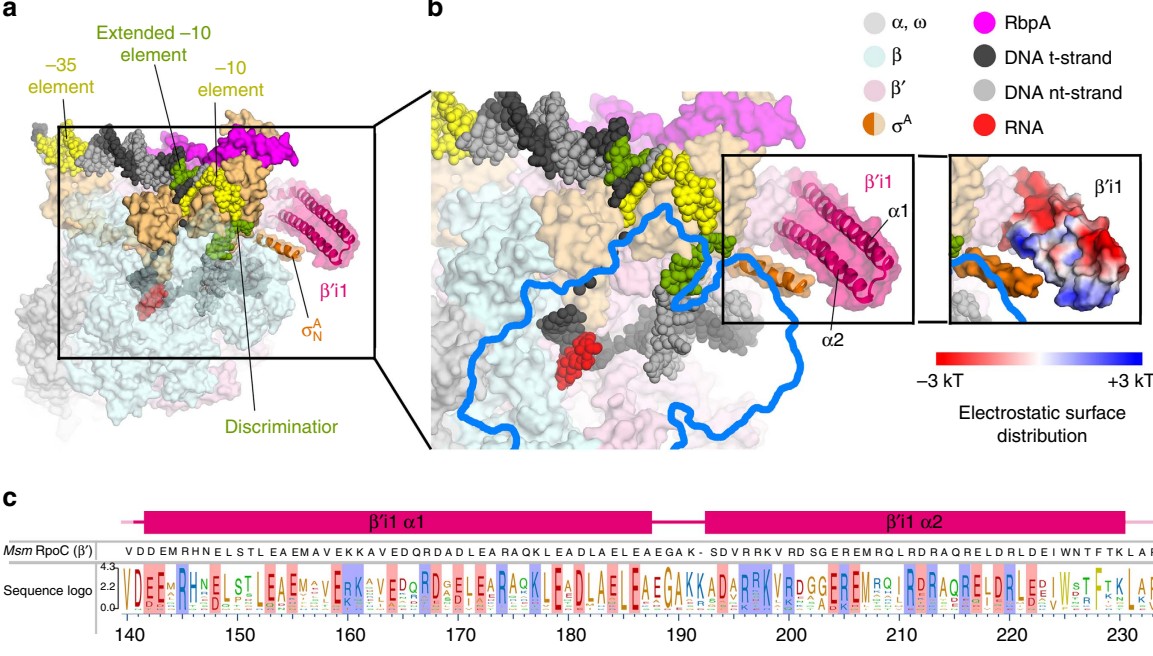

**Figure 4 | The Actinobacteria β′i1. (a)** Overall view of the *Msm* RbpA/RPo. The color-coding of most of the structural features is denoted in the legend (Fig. 3b). Protein components (core RNAP, σ$^A$, RbpA) are shown as molecular surfaces. The surface of the RNAP β subunit is transparent, revealing the nucleic acids and σ$^A$ elements located in the RNAP active centre cleft. The surfaces of σ$^A_N$ and the lineage-specific insert β′i1 are transparent, revealing the α-carbon backbone ribbons underneath. The nucleic acids are shown as CPK atoms. The boxed region is magnified in Fig. 3b. **(b)** Magnified view of the boxed region from Fig. 3a. Obscuring elements of the RNAP β subunit have been removed (outlined in blue), revealing the nucleic acids in the RNAP active site cleft. The boxed region shows β′i1, with the two α-helices (α1, α2) labelled. The boxed region on the right shows the same view of β′i1 but with the molecular surface coloured according to the electrostatic surface potential (red, −3 kT; blue, +3 kT)[66], illustrating the asymmetric charge distribution. **(c)** The secondary structure of the *Msm* β′i1 (*Msm* β′ residues 141–230, *Msm* numbering shown at the bottom) is schematically illustrated (α-helices are shown as rectangles), with the *Msm* β′ sequence shown below. The sequence logo[67] shown below was derived from a sequence alignment of 720 *Msm* β′i1 homologues (all Actinobacteria). Conserved negatively charged (D/E) and positively charged (K/R) positions on the α-helices are shaded red or blue, respectively. The α1 harbours 14 positions of conserved negative charge and 6 positions of conserved positive charge (net charge −8), while α2 harbours 7 positions of conserved negative charge and 9 positions of conserved positive charge (net charge +2).

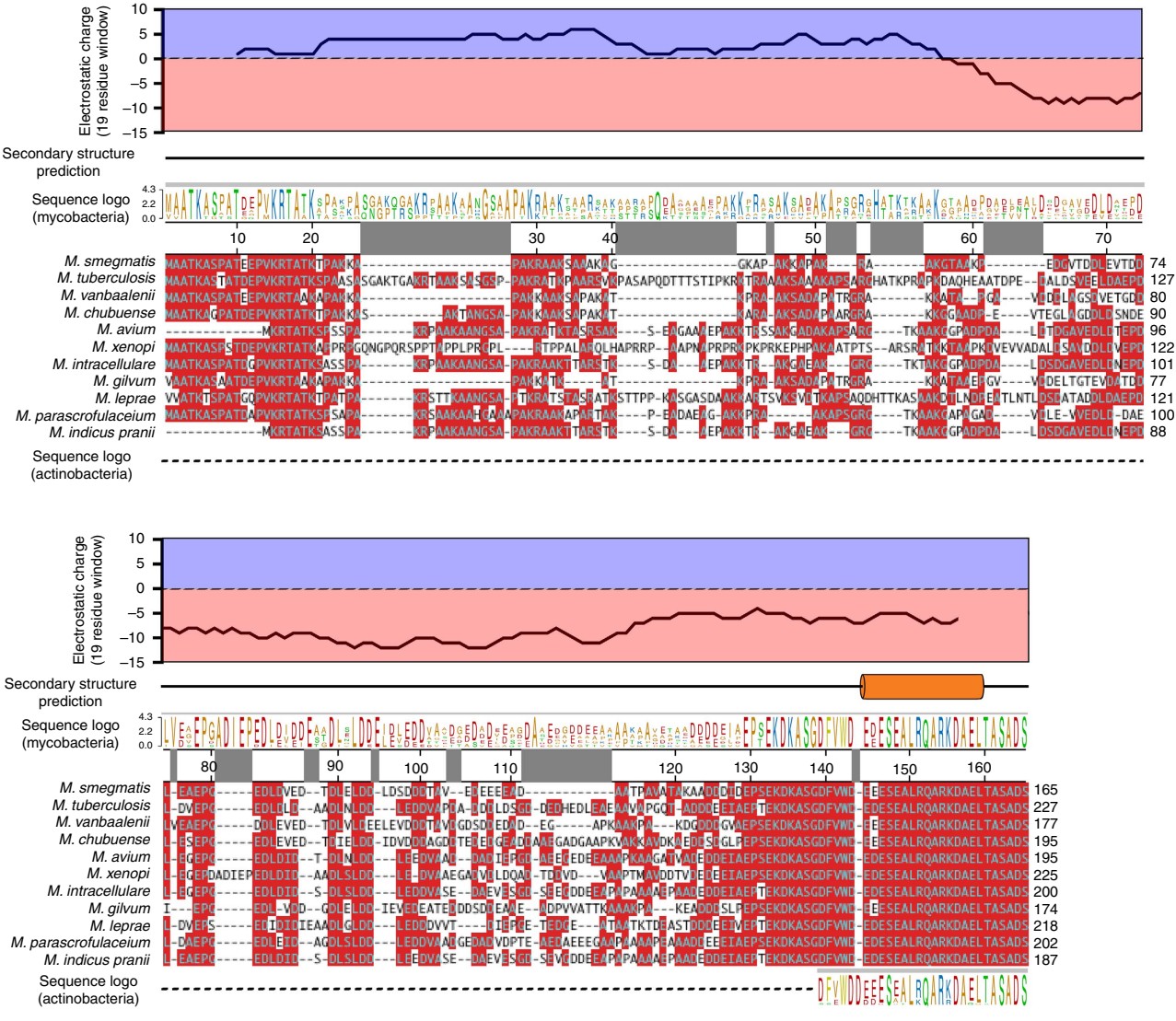

**Figure 5 | Sequence characteristics of the σ$^A_N$ of mycobacteria and Actinobacteria.** The *Msm* σ$^A_N$ (163 residues N-terminal of conserved region 1.2 (ref. 68)) is aligned with representative mycobacteria σ$^A_N$ sequences (the number scale on top of the alignment shows the *Msm* σ$^A$ numbering). Residues conserved in more than half of the sequences are shaded red. The sequence logo[67] shown above was derived from an alignment containing 45 mycobacteria σ$^A_N$ sequences. The predicted secondary structure[48,49] for the *Msm* σ$^A_N$ is shown schematically above the logo. One α-helix is predicted (*Msm* σ$^A_N$ residues 144–159, orange cylinder), with the rest of the sequence lacking any secondary structure. The net electrostatic charge of the derived consensus sequence, calculated in a 19-residue window, is plotted at the top (net positive charge is shaded blue, negative charge shaded red). A sequence logo derived from the *Msm* σ$^A_N$ with 199 Actinobacteria σ$^A_N$ sequences (excluding other mycobacteria sequences) is shown below. The region corresponding to *Msm* σ$^A_N$ residues 139–165, which includes the predicted α-helix, was conserved with all Actinobacteria σ$^A_N$ sequences, while *Msm* σ$^A_N$ residues 1–138 (the region predicted to lack secondary structure) showed no sequence relationship with other Actinobacteria σ$^A_N$ sequences (except for other mycobacteria σ$^A_N$ sequences) and could not be aligned.

sequence bears no apparent relationship with Group 1 σ$_N$ sequences from several clades, including ε-proteobacteria, deinococcus-thermus (Supplementary Fig. 7C), Mollicutes, Actinobacteria (Fig. 5), Cyanobacteria, Bacteroidetes and Chlorobi (Supplementary Table 2).

The *Msm* TIC structures contain full-length σ$^A$, and clear electron density for an α-helix extending from the N-terminus of σ$^A_{1.2}$ was observed (Fig. 6a; Supplementary Fig. 8). We have not been able to unambiguously assign the sequence register of the σ$^A$ N-terminal helix. *Msm* σ$^A$ harbours 163 residues N-terminal to σ$^A_{1.2}$ (σ$^A_N$). Secondary structure prediction algorithms predict a total lack of secondary structure for residues 1–143 (refs 48,49), and one α-helix is predicted (~ residues 145–160) as observed in our structure (Figs 5

and 6a). Strikingly, Blast searches identified sequence homologues for *Msm* σ$^A_N$ only among Actinobacteria, and for most Actinobacteria, the only conserved sequence segment corresponded to *Msm* σ$^A$ residues 139–163, corresponding to the α-helix observed in the structure (Figs 5 and 6a). The mycobacteria σ$^A_N$ displays a striking separation of charge, where roughly the first half of the sequence is biased towards positive charged residues, while the C-terminal half is biased towards negatively charged residues (Fig. 5).

The orientation of the *Msm* σ$^A_N$-helix positioned between the RNAP β2 domain and β'i1, places the rest of the *Msm* σ$^A_N$ outside of the RNAP active site cleft near the β2 domain (Fig. 6). The placement of the *Msm* σ$^A$ N-terminal helix is suggestive that it defines the path of *Eco* σ$^{70}_{1.1}$ after being displaced from the

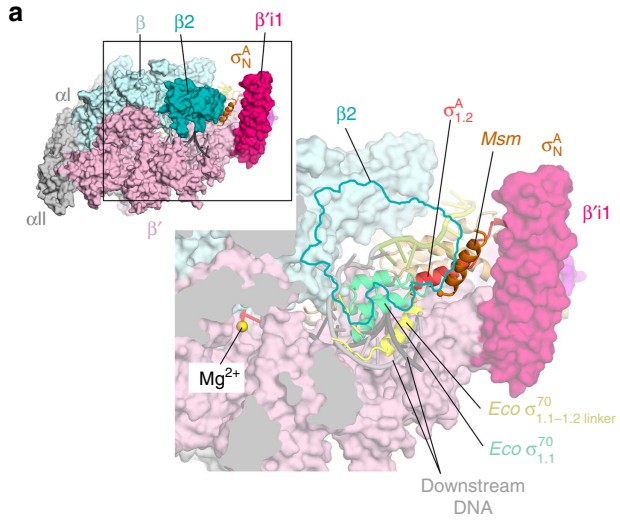

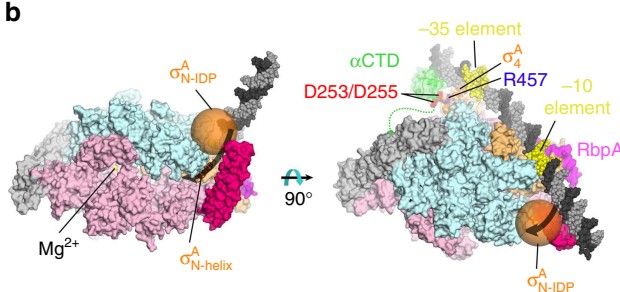

**Figure 6 | Structural and functional context of the *Msm* $\sigma^A_N$.** (**a**) (upper left) Overall view of the *Msm* RbpA/RPo. The color-coding is the same as Fig. 1b except the β2 domain is coloured slate blue. The RNAP is shown as a molecular surface except $\sigma^A$ is shown as a backbone ribbon. The nucleic acids are shown in cartoon format. The boxed region is magnified on the lower right. (lower right) Magnified view of the boxed region from the upper right. The β2 domain (outlined in blue) as well as other obscuring regions of the structure have been removed, revealing the RNAP active site cleft and the nucleic acids therein. The $\sigma^A$, shown as a backbone ribbon, is coloured light orange, except the $\sigma^A_{1.2}$ N-terminal helix is coloured orange-red, and the $\sigma^A_N$-α-helix is coloured orange. Shown in the downstream duplex DNA-binding channel is the superimposed position of *Eco* $\sigma^{70}_{1.1}$ (light green) and the $\sigma^{70}_{1.1-1.2}$ linker (yellow) from PDB ID 4LK1 (ref. 6). (**b**) Model of an *Msm* closed promoter complex (RPc or RP1)[4] summarizing the role of the αCTD (green), β'i1 (hot pink), and $\sigma^A_N$ (orange) in RPo formation. The αCTD was modelled bound upstream of the −35 element, placing conserved negatively charged αCTD residues D253/D255 (Fig. 2e; shown as red CPK atoms) near conserved $\sigma^A_4$ R457 (blue CPK atoms), inferring interactions analogous to *Eco* CTD D259/E261 with $\sigma^{70}_4$ R603 (ref. 26). The $\sigma^A_{N-helix}$ is shown as an orange backbone ribbon as observed in the crystal structures (Fig. 5a). The rest of the $\sigma^A_N$ (approximately residues 1–143) is predicted to comprise an intrinsically disordered region (IDR) and is modelled as a (transparent orange) sphere with the expected radius (30 Å) of a Flory random coil for an IDR with κ (measurement of the extent of charge separation) of 0.42 (ref. 50). The sphere was placed to connect with the N-terminus of the $\sigma^A_{N-helix}$ and simultaneously minimize steric clashes. The combined placement of β'i1 and the $\sigma^A_{N-IDR}$ blocks the path the DNA must traverse to enter the active site cleft (indicated by thick black arrows).

RNAP active site cleft by incoming promoter DNA[6,47]. However, it is not clear that movements of mycobacteria $\sigma^A_N$ during transcription initiation will parallel those of *Eco* $\sigma^{70}_{1.1}$ for the following reasons:

(1) *Msm* $\sigma^A_N$ bears no sequence nor structural relationship with $\sigma^{70}_N$ of *Eco* $\sigma^{70}$ (Fig. 5, Supplementary Fig. 7; Supplementary Table 2) and therefore cannot be assumed to bear a functional relationship.

(2) *Msm* $\sigma^A$[1–143] is predicted to behave like an intrinsically disordered region and, as such, is predicted to have a molecular volume much too large to fit in the RNAP active site cleft[50].

(3) In the *Msm* RbpA/TIC structures, $\sigma^A_N$ is located outside of the RNAP active site cleft whether or not nucleic acids are present in the active site cleft.

This analysis suggests that unlike *Eco* $\sigma^{70}_{1.1}$, *Msm* $\sigma^A_N$ may never reside in the RNAP active site cleft. Like *Eco* $\sigma^{70}_{1.1}$, however, the placement of *Msm* $\sigma^A_N$ between the RNAP β2 domain and β'i1 suggests that *Msm* $\sigma^A_N$ plays a role in regulating RPo formation by blocking or restricting the entrance of the DNA template into the active site cleft (Fig. 6b)[45,46].

## Discussion

Here we present in-depth analyses leading to a comprehensive summary of the structural and functional features of mycobacteria RNAP that are similar to and distinct from *Eco*. The high resolution views of *Msm* E$\sigma^A$/promoter DNA interactions detailed here (Fig. 1, Supplementary Figs 2–4) are conserved with those seen in *Thermus* but with new interactions observed due to the increased resolution from previous *Thermus*[16,37,41,51] and *Eco* (ref. 52) structures. Our results point to a conserved role for αCTD/UP element DNA interactions in regulating transcription initiation between *Eco* (ref. 22) and mycobacteria (Fig. 3, Supplementary Fig. 5), a role not shared with *Thermus* (Fig. 3e,f). Although the insertion point of a lineage-specific insert is identical in the β' subunit of *Thermus* and Actinobacteria RNAP (Supplementary Fig. 6), the structures and likely functional roles of the inserts in their respective organisms are unrelated (and the insert is absent in *Eco* RNAP). Finally, the Group 1 σ factors are among the most highly conserved proteins across all of the bacterial kingdom[53]. Nevertheless, a defining feature of Group 1 σ's, $\sigma_N$ that harbours *Eco* $\sigma^{70}_{1.1}$, bears no apparent structural relationship with the $\sigma^A$ N-terminal extensions for either *Thermus* or Actinobactera (Figs 5 and 6a, Supplementary Fig. 7, Supplementary Table 2). We postulate that the Actinobacteria $\sigma^A_N$ plays a similar functional role as *Eco* $\sigma^{70}_{1.1}$ (licensing access of nucleic acids to the active site channel), but does so in a mechanistically unique way. This work illustrates that structural and functional paradigms developed from the study of the *Eco* transcription system are not universally applicable among bacteria, and highlights the importance of studying phlyogenetically distinct bacteria to gain comprehensive insight into transcription and its regulation.

Paradigms of transcription have evolved from years of studies using *Eco* as the model organism. Investigations of transcription mechanisms in other bacteria continue to reveal significant differences with mechanisms in *Eco*. For example, although the overall mechanism of RPo formation appears to be conserved[4], mycobacteria have unique activators that act at the ds/ss (−12/−11) junction instead of at canonical sites upstream of the −35 element[4]. Why? We propose that this positioning is critical to overcome the block created by the combination of the *Msm* $\sigma^A_N$ and β'i1 (Fig. 6b): CarD and RbpA stabilize the bend that places DNA into the RNAP active site cleft[4]. Lacking this reinforcement, mycobacteria RPo readily dissociates, presumably because DNA loses the competition for active site cleft access. In contrast, no lineage-specific insertions in *Eco* directly block the active site[15,54]. Instead two insertions absent in Actinobacteria

may form a stabilizing clamp on the downstream duplex DNA, creating a highly stable RPo at many promoters[3].

During the review of this manusript, a report describing a number of crystal structures (ranging between 4.3 and 3.8 Å resolution) of $Mtb$ E$\sigma^A$ bound to a downstream-fork (ds-fork) promoter fragment was published[55]. The $Mtb$ E$\sigma^A$/ds-fork complex is remarkably similar to the $Msm$ RPo: superimposing the $Mtb$ E$\sigma^A$/ds-fork complex (PDB ID 5UHA)[55] with the $Msm$ RPo results in a r.m.s.d. of 1.02 Å over 2,783 α-carbon positions. The crystal packing environments of the $Mtb$ (space group $P2_12_12_1$) and $Msm$ (space group $P2_1$) structures are very similar.

The $Mtb$ E$\sigma^A$/ds-fork structures revealed the Actinobacteria-specific β′i1 and the $\sigma^A_{N-helix}$ in essentially identical conformations to those we observe in the $Msm$ TIC structures (Figs 4a,b and 6a)[4] but Lin et al.[55] proposed a different mechanistic role for these structural elements. Lin et al.[55] suggest that in the $Mtb$ E$\sigma^A$, the $\sigma^A$ N-terminal extension resides in the RNAP active site cleft and is displaced by the entering promoter DNA in a manner analagous to Eco $\sigma^{70}_{1.1}$ (refs 6,47). Once ejected from the RNAP active site cleft, the $Mtb$ $\sigma^A_N$ is proposed to cooperate with β′i1 to trap the promoter DNA in the cleft, thereby stabilizing RPo. In support of this hypothesis, Lin et al.[55] report that deleting $\sigma^A_N$, β′i1, or both destabilizes complexes of the resulting holoenzymes with the ds-fork DNA.

However, unlike the compact folded domain Eco $\sigma^{70}_{1.1}$ (ref. 6), mycobacterial $\sigma^A_N$'s are predicted to be intrinsically disordered regions (IDRs; Fig. 5). The IDRs of the $Msm$ (143 residues) and $Mtb$ (205 residues) $\sigma^A_N$'s (Fig. 5) are predicted to have effective molecular radii of $\sim 30$ and 34 Å, respectively (Fig. 6b)[50], too large to fit into the RNAP cleft. So while Eco $\sigma^{70}_{1.1}$ ($\sim 90$ residues, molecular radius of $\sim 14$ Å) starts in the RNAP cleft[6], we propose that the mycobacterial $\sigma^A_N$ initially lies outside the cleft. In our model of E$\sigma^A$, the mycobacterial $\sigma^A_N$ with β′i1 cooperate to impede promoter DNA entry into the channel (Fig. 6b) but do not exclude the possibility that one or both inserts could stabilize the DNA once established in the RNAP cleft as proposed by Lin et al.[55].

The ds-fork promoter template studied by Lin et al.[55] is 'pre-melted' and lacks DNA upstream of its single-stranded −10 element. The association pathway of this template is unknown but is almost certainly very different than the assocation pathway of a fully duplex promoter. Initial recognition of a fully duplex promoter occurs outside of the RNAP cleft, with subsequent initiation of −10 element melting (also outside of the RNAP cleft), bending of the downstream DNA across the entrance to the cleft, and finally full transcription bubble melting and loading of the DNA into the cleft (Fig. 6b)[3,4]. Thus, we suggest that the effect of deleting $Mtb$ $\sigma^A_N$ and/or β′i1 in stabilizing ds-fork binding[55] may not reflect the effects of these deletions on fully duplex promoter interactions. In summary, we propose a very different role for the mycobacteria $\sigma^A_N$ from Lin et al.[55], one where $\sigma^A_N$ is never in the RNAP cleft and cooperates with β′i1 to impede DNA entry into the cleft. Clearly, further studies are required to understand the role of the mycobacterial $\sigma^A_N$ and β′i1 in both the association and dissociation of fully duplex promoter DNA and how these elements are regulated.

Continued emergence of multi-antibiotic resistant bacteria present the sobering reality that the clinical anti-bacterial arsenal is becoming increasingly depleted[56]. We suggest studies of phlyogenetically distinct bacteria such as mycobacteria provide the fudamental groundwork needed to develop novel antibiotics to combat TB and other devastating bacterial diseases.

## Methods

**Protein expression and purification.** Msm $RbpA/\sigma^A$. $Msm$ pET-SUMO $\sigma^A$ and pET21c-RbpA were co-expressed in $Eco$ BL21 (DE3) by induction with 0.5 mM isopropyl-beta-D-thiogalactopyranoside (IPTG) for 3 h at 30 °C, affinity purified on a Ni$^{2+}$-column, and cleaved by ULP1 protease overnight[4]. The cleaved complex was loaded onto a second nickel column, collected from the flow-through and further purified by size exclusion chromatography (Superdex 200, GE Healthcare).

$Msm$ RNAP. $Msm$ RNAP was purified from the $Msm$ mc2155 strain expressing a native chromosomal copy of rpoC with a C-terminal His$_{10}$-tag[4]. $Msm$ cells were grown at the Bioexpression and Fermentation Facility at the University of Georgia, lysed, and core RNAP was precipitated by polyethyleneimine (PEI) precipitation (0.35% w/v). Protein was precipitated with ammonium sulfate (35% w/v) and purified on a Ni$^{2+}$-affinity column. Fractions containing RNAP were loaded on a Biorex (BioRad) column and RNAP was eluted with a salt gradient. A five-fold molar excess of the purified $Msm$ RbpA/$\sigma^A$ was added to the core RNAP and the resulting holoenzyme was further purified by size exclusion chromatography. The purified complex was dialyzed into 20 mM Tris-HCl, pH 8, 100 mM K-glutamate, 10 mM MgCl$_2$, 1 mM DTT, concentrated by centrifugal filtration to $\sim 15$ mg ml$^{-1}$, and stored at $-80$ °C.

$Eco$ RNAP. $Eco$ core RNAP was overexpressed and purified from $Eco$ BL21(DE3) cells co-transformed with pEcrpoABC(-XH)Z (encoding $Eco$ RNAP rpoA, rpoB and rpoC-His$_{10}$) and pACYCDuet-1_Ec_rpoZ (encoding rpoZ)[57]. $Eco$ RNAP subunits were co-overexpressed overnight at room temperature for $\sim 16$ h after induction with 0.1 mM IPTG. Cells were lysed, and core RNAP was precipitated with 0.6% PEI. Proteins eluted from the PEI pellet were then purified by Ni$^{2+}$-affinity chromatography, Bio-Rex 70 chromatography, and finally purified by size exclusion chromatography[4,10,14]. $Eco$ core RNAP (Δ-αCTD) was purified as described for the full-length protein but using protein expressed from pECrpoA(-X$_{234-241}$H)BCZ containing a PreScission protease site between the αNTD and αCTD-His$_{10}$ (ref. 57). The only difference in the purification occurred after the first Ni$^{2+}$-affinity step where the protein was subjected to PreScission protease cleavage, dialyzed to remove imidazole. The sample was reapplied to a Ni$^{2+}$-affinity column and the flow-through collected for subsequent steps[57].

$Eco$ $\sigma^{70}$ was expressed from a pET21a-based expression vector encoding an N-terminal His$_6$-tag followed by a PreScission protease (GE Healthcare) cleavage site. The protein was expressed using standard methods and purified by Ni$^{2+}$-affinity chromatography, protease cleavage to remove the His$_6$-tag, anion exchange chromatography, and finally size exclusion chromatography.

Mtb RNAP. $Mtb$ core RNAP subunits were co-overexpressed in $Eco$ BL21 (DE3) pRARE2 (Novagen) cells overnight at room temperature for $\sim 16$ h after induction with 0.1 mM IPTG[10]. Cells were lysed, and core RNAP was precipitated with 0.6% PEI. Proteins eluted from the PEI pellet were then purified by Ni$^{2+}$-affinity chromatography and subsequently purified by size exclusion chromatography[4,10,14].

$Mtb$ $\sigma^A$ was expressed from pAC27 (ref. 9) in $Eco$ BL21 (DE3) pRARE2 and purified by Ni$^{2+}$-affinity chromatography and size exclusion chromatography[9].

**Crystallization of Msm RbpA/RPo.** To generate the full $Msm$ RPo, $Msm$ RbpA/ E$\sigma^A$ was mixed in a 1:1 molar ratio with duplex promoter DNA scaffold ($-37$ to $+13$) and a five-fold molar excess of RNA primer complementary to the t-strand DNA from $+1$ to $-3$ (GE Dharmacon, Lafayette, CO, United States; Fig. 1a) as previously described Bae:2015fc}. Crystals were grown by hanging drop vapour diffusion by mixing 1 μl of $Msm$ RbpA/RPo solution (11 mg ml$^{-1}$ protein) with 1 μl of crystallization solution [0.1 M Bis–Tris, pH 6.0, 0.2 M LiSO$_4$, 16% (w/v) polyethylene glycol 3350, 2.5% (v/v) ethylene glycol] and incubating over a well containing crystallization solution at 22 °C. The crystals were cryo-protected by step-wise transfer (three steps) into 0.1 M Bis–Tris, pH 6.0, 0.2 M LiSO$_4$, 22% (w/v) polyethylene glycol 3350, 20% (v/v) ethylene glycol and flash frozen by plunging into liquid nitrogen.

**Data collection and structure determination.** X-ray diffraction data were collected at the Argonne National Laboratory Advanced Photon Source (APS) NE-CAT beamline 24-ID-E at a wavelength of 0.97918 Å. Structural biology software was accessed through the SBGrid consortium[58]. Data were integrated and scaled using HKL2000 (ref. 59).

An initial electron density map was calculated by molecular replacement using Phaser[60] from a starting model of the $Msm$ RbpA/E$\sigma^A$/us-fork structure (PDB ID 5TW1)[4]. The model was first improved using rigid body refinement of 20 individual mobile domains using PHENIX[61]. The resulting model was improved by iterative cycles of manual building with COOT[62] and refinement with PHENIX[61]. The final refined model had 94% of residues in the favored region of the Ramachandran plot, 0.71% in the region of Ramachandran outliers.

**In vitro transcription assays.** *In vitro* abortive initiation transcription assays were performed at 37 °C as described[10], or at 50 °C with $Tth$ holo. VapB and VapBUP promoter templates were prepared using PCR amplification on a synthesized template (Integrated DNA Technologies; Supplementary Fig. 6). Assays with VapB and VapBUP promoter templates were performed in assay buffer (10 mM Tris-HCl, pH 8.0, 10 mM MgCl$_2$, 0.1 mM EDTA, 0.1 mM DTT, 50 μg ml$^{-1}$ BSA) with 100 mM K-glutamate when using $Msm$ holoenzyme, or with 50 mM K-glutamate with $Mtb$ holoenzyme. Assays with $Eco$ RNAP and $Eco$ ΔαCTD-RNAP were performed in assay buffer with 50 mM KCl.

Abortive initiation assays were initiated on VapB and VapBUP templates with ApU dinucleotide primer (250 μM; Trilink Biotechnologies, San Diego, CA), [α-$^{32}$P]GTP (1.25 μCi; Perkin Elmer Life Sciences, Waltham, MA), and unlabelled GTP (50 μM; GE Healthcare Life Sciences). *Mbo, Msm, Tth* RNAPs (50 nM) were added to DNA template (10 nM) and nucleotide mix, and the reaction was incubated for 10 min at 37 °C (*Mbo* or *Msm*) or 50 °C (*Tth*). *Eco* holoenzyme or *Eco* ΔαCTD-holoenzyme were used at 5 nM and the DNA template was 1 nM. Transcription products were visualized by polyacrylamide gel electrophoresis (23%) followed by phosphorimagery and quantitation using Image J.

**Promoter searches and annotation.** To annotate and compare promoter motifs we performed identical analyses on *Eco* and *Mtb*, using previously determined TSS from each organism. The 3,746 TSSs (Table 1) from *Eco* were previously determined in *Eco* K-12 MG1655 cells growing in mid-exponential phase[27]. The 1,779 TSSs (Table 1) from *Mtb* were previously determined from exponentially growing *Mtb* H37Rv (ref. 28).

Using defined spacings from the −10 promoter to the TSS[30] and the sequence characteristics and variable spacing between the promoter elements defined in *Eco*[31], we searched the deposited RNA-seq sequences within 50 bp upstream of the identified TSSs for motifs as denoted in Table 1. We accounted for the variability in spacing between the −10 element and the TSS[30] and between the −10 and −35 elements by performing separate searches[31]. Searches were performed using Microsoft Excel Filter functions.

**Data availability.** The original 2.76 Å resolution *Msm* RbpA/Eσ$^A$/us-fork coordinates (5TW1)[4] have been updated by the addition of some water molecules, including the ordered water facilitating αCTD binding to DNA (Fig. 3c,d) and have been deposited in the Protein Data Bank with accession ID 5VI8. The X-ray crystallographic coordinates and structure factor file for the *Msm* RbpA/RPo structure have been deposited in the Protein Data Bank with accession ID 5VI5. The data that support the findings of this study are available from the corresponding author upon request.

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

# ARTICLE

46. Vuthoori, S., Bowers, C. W., McCracken, A., Dombroski, A. J. & Hinton, D. M. Domain 1.1 of the σ70 subunit of *Escherichia coli* RNA polymerase modulates the formation of stable polymerase/promoter complexes. *J. Mol. Biol.* **309,** 561–572 (2001).

47. Mekler, V. *et al.* Structural organization of bacterial RNA polymerase holoenzyme and the RNA polymerase-promoter open complex. *Cell* **108,** 599–614 (2002).

48. Rost, B., Yachdav, G. & Liu, J. The PredictProtein server. *Nucleic Acids Res.* **32,** W321–W326 (2004).

49. Linding, R. *et al.* Protein disorder prediction: implications for structural proteomics. *Structure* **11,** 1453–1459 (2003).

50. Das, R. K. & Pappu, R. V. Conformations of intrinsically disordered proteins are influenced by linear sequence distributions of oppositely charged residues. *Proc. Natl Acad. Sci. USA* **110,** 13392–13397 (2013).

51. Zhang, Y. *et al.* Structural basis of transcription initiation. *Science* **338,** 1076–1080 (2012).

52. Zuo, Y. & Steitz, T. A. Crystal structures of the *E. coli* transcription initiation complexes with a complete bubble. *Mol. Cell* **58,** 534–540 (2015).

53. Gruber, T. M. & Bryant, D. A. Molecular systematic studies of eubacteria, using sigma70-type sigma factors of group 1 and group 2. *J. Bacteriol.* **179,** 1734–1747 (1997).

54. Opalka, N. *et al.* Complete structural model of *Escherichia coli* RNA polymerase from a hybrid approach. *PLoS Biol.* **8,** e1000483 (2010).

55. Lin, W. *et al.* Structural basis of *Mycobacterium tuberculosis* transcription and transcription inhibition. *Mol. Cell* **66,** 169–179 (2017).

56. World Health Organization. Global priority list of antibiotic-resistant bacteria to guide research, discovery, and development of new antibiotics. http://www.who.int/medicines/publications/global-priority-list-antibiotic-resistant-bacteria/en/ (2017).

57. Twist, K.-A. *et al.* A novel method for the production of *in vivo*-assembled, recombinant *Escherichia coli* RNA polymerase lacking the α C-terminal domain. *Protein Sci.* **20,** 986–995 (2011).

58. Morin, A. *et al.* Collaboration gets the most out of software. *Elife* **2,** e01456 (2013).

59. Otwinowski, Z. & Minor, W. Processing of X-ray diffraction data collected in oscillation mode. *Methods Enzymol.* **267,** 307–326 (1997).

60. McCoy, A. J. *et al.* Phaser crystallographic software. *J. Appl. Crystallogr.* **40,** 658–674 (2007).

61. Adams, P. D. *et al.* PHENIX: a comprehensive Python-based system for macromolecular structure solution. *Acta Crystallogr. D Biol. Crystallogr.* **66,** 213–221 (2010).

62. Emsley, P. & Cowtan, K. Coot: model-building tools for molecular graphics. *Acta Crystallogr. D Biol. Crystallogr.* **60,** 2126–2132 (2004).

63. Gaal, T. *et al.* Promoter recognition and discrimination by EsigmaS RNA polymerase. *Mol. Microbiol.* **42,** 939–954 (2001).

64. Feklistov, A. *et al.* A basal promoter element recognized by free RNA polymerase σ Subunit determines promoter recognition by RNA polymerase holoenzyme. *Mol. Cell* **23,** 97–107 (2006).

65. Haugen, S. P. *et al.* rRNA promoter regulation by nonoptimal binding of σ region 1.2: an additional recognition element for RNA polymerase. *Cell* **125,** 1069–1082 (2006).

66. Baker, N. A., Sept, D., Joseph, S., Holst, M. J. & McCammon, J. A. Electrostatics of nanosystems: application to microtubules and the ribosome. *Proc. Natl Acad. Sci. USA* **98,** 10037–10041 (2001).

67. Schneider, T. D. & Stephens, R. M. Sequence logos: a new way to display consensus sequences. *Nucleic Acids Res.* **18,** 6097–6100 (1990).

68. Lonetto, M., Gribskov, M. & Gross, C. A. The sigma 70 family: sequence conservation and evolutionary relationships. *J. Bacteriol.* **174,** 3843–3849 (1992).

## Acknowledgements

We thank R. Saecker for insightful discussion. This work is based upon research conducted at the Northeastern Collaborative Access Team beamlines, which are funded by the NIGMS from the NIH (P41 GM103403). This research used resources of the Advanced Photon Source, a U.S. Department of Energy (DOE) Office of Science User Facility operated for the DOE Office of Science by Argonne National Laboratory under Contract No. DE-AC02-06CH11357. The use of the Rockefeller University Structural Biology Resource Center was made pssible by NIH/NCRR 1S10RR027037. This work was supported by NIH grant RO1 GM114450 to E.A.C.

## Author contributions

All authors designed the project, performed the experiments, and analysed the data. S.A.D. and E.A.C. wrote the paper.

## Additional information

**Competing interests:** The authors declare no competing financial interests.

