## [Peer Review File · Nature Communications]

Reviewers' Comments:

Reviewer #1 (Remarks to the Author):

This manuscript reports a structure at 3.2Å of an *M. smegmatis* (Msm) initiation complex that extends from -37 to +13 with respect to the transcription start site and contains a promoter bubble and a short RNA “product”. The paper also contains further analysis of a previously published fork junction complex at 2.76Å. The 3.2Å structure is at higher resolution than any other initiation complex published previously, and it provides some useful new details about this complex. A major emphasis of the paper is an adventitious alpha CTD-DNA interaction in the fork-junction complex structure that mimics the natural alpha CTD interaction with UP element DNA in *E. coli* promoters. This interaction is between the alpha CTD in one RNAP with an A-rich DNA sequence downstream of the -35 hexamer bound to a neighboring symmetry-related RNAP. Although this interaction is not physiologically relevant, the actual details of the protein-DNA contacts are essentially identical to those in complexes with natural UP elements in *E. coli* transcription complexes, leading the authors to test whether there are natural UP elements in Msm. The authors explore the significance of UP element interactions by performing transcription and bioinformatic analyses that ultimately demonstrate the occurrence and functional importance of UP elements in mycobacterial promoters. The authors also report the structure of two lineage-specific parts of RNAP unique to Actinobacteria, an insert in beta prime and an N-terminal extension in the major sigma factor. I have only minor suggestions. There is plenty here to justify publication.

1. The Ebricht group just published a structure of a *Mycobacterium tuberculosis* crystal structure of an open complex, Lin et al. 2017 Mol Cell. Although the emphases of the two papers are very different, two subjects that are discussed in both papers should be compared/contrasted. Specifically, it would be helpful to be more explicit about differences in the interpretations of the beta prime insertion as a gate and about the relevance of the N-terminal domain of sigma in Mtb to sigma 1.1.

2. If I understand correctly, the Msm alphaCTD-DNA interaction described here is actually from the structure published previously, i.e. with the scaffold shown in Fig. S1 and Fig. 2, not with the scaffold shown in Fig. 1. The previous paper focused on the importance of a transcription factor and on the kinetic mechanism, but did not mention at all on the adventitious interaction between the promoter DNA just downstream of the -35 hexamer in one RNAP complex with the aCTD in the adjacent RNAP. The phrasing on page 6 is accurate, but it is sparse and on first reading it is unclear whether this is a new structure or the one published previously. It should be emphasized more clearly that this is not the physiologically relevant position for the aCTD-UP element interaction. On line 130, it should also be stated more clearly that D259 and E261 in aCTD and sigma R603 from *E. coli* RNAP are all conserved with the analogous residues in Msm RNAP and that if aCTD were bound to an UP element-like sequence upstream of the -35, these Msm residues would likely interact. However, in the adventitious interaction

this cannot happen because sigma R603 is too far away. This situation doesn't really become clear until Fig. 5B.

3. Line 21. Define Msm.

4. Line 140. The UP element bioinformatic analysis appears to have been done for just a proximal subsite. Was it also done with a distal subsite (i.e. a binding site for the 2nd aCTD)?

5. Line 301. While many E. coli transcription experts may have worked under the assumption that the E. coli paradigm would apply to all bacteria, this sentence should be rephrased. It does a disservice to the many labs who long ago started studying transcription in other bacterial systems with just the opposite assumption, i.e. that there would be major differences. Given the 3 billion years of evolution since divergence of gram positives and gram negatives, for example, it is surprising how much similarity has survived!

6. References 26 and 56 in the bibliography are incomplete.

7. Line 696. Since the sigma-aCTD contact does not actually occur in this complex, it should be made clearer that this interaction is inferred from the E. coli situation.

8. Line 765: "analogous"

Reviewer #2 (Remarks to the Author):

The manuscript by Hubin et al. mainly focuses on the structural aspects of the mycobacteria transcription initiation complex (TIC). The primary content is based on a newly solved 3.2-angstrom crystal structure of an RNA polymerase (RNAP) open promoter complex (RPO) AND a previously reported 2.8-angstrom crystal structure of an RNAP-upstream fork DNA complex, both from Msm (*M. smegmatis*). These are the pioneer structures of mycobacteria RNAP and will benefit the understandings of related mycobacteria RNAPs and anti-mycobacteria studies. These two structures, solved at higher resolution than previously reported bacteria TIC structures, confirm the overall architecture and the protein:nucleic acid interaction framework of the complex, and provide better details. More importantly, this manuscript highlights a few previously unclarified features of the Msm RNAP: 1. The possible regulatory role of the alpha subunit CTD (a-CTD) through its interactions with the UP element; 2. The lineage-specific beta' subunit insertion (b'11) and its possible contribution to mycobacteria transcription initiation; 3. The unique features of the Actinobacteria-specific sigma A N-terminal region (SigA-N) and its disposition in the context of TIC. While the a-CTD function is supported also from sequence conservation analysis and in vitro transcription data, the mechanisms underlying the b'11 and SigA-N remain to be clarified.

Comments and concerns:

1. Because the Msm fork junction-structure (pdb: 5TW1) is of higher resolution and is described in detail, in particular for the interactions around the upstream edge of the TIC transcription bubble. The authors need to make it clearer that their previously published paper (ELife 2017) focuses more on the RbpA/CarD topic, while other important features of this structure is described for the first time in this manuscript.

2. As the Msm fork-junction structure and the RPo structure may represent different critical stages of transcription initiation, it will be of great interest to describe the notable conformational differences between these two structures both in text and figure formats.

3. The nucleic acids conformation in the Msm RPo complex largely resembles that in Thermus RPo structure (pdb: 4XLN), and the primary difference is the disorder of the -11 to -7 region of the template (t) strand in the Msm structure. The interactions between this region and the RNAP may contribute to the maintenance of the open complex and are likely related to TIC bubble collapse as transcription progresses from initiation to elongation. Could the authors comment on possible reasons that may lead to the disorder of this t-strand region in the Msm structure, or any implications related to the disorder?

4. In the crystallography-related statistics (supplementary Table 1), Rmerge and I/sigI values (1.751 and 0.63) of the highest resolution shell do not meet general criteria. It is understandable that the authors want to push the resolution limit a bit for better observation of structural details. However, it will be then necessary to provide omit-type electron density maps to aid the appreciation of structural details, in particular in regions with novel information. 2Fo-Fc maps are provided in the supplementary materials mainly for the fork-junction structure but not for the newly reported RPo structure.

5. It is quite interesting that the interactions between the alpha-CTD and the promoter DNA region -29 to -24 are observed from a crystal packing-derived event. The authors performed sequence analysis and functional assays to reveal the possibility of transcription regulation by mycobacteria alpha-CTD through its interactions with the UP element (upstream of -30 region), and generated a structural model with the alpha-CTD interacting with the UP element. These are great efforts toward the understanding of the mycobacteria alpha-CTD function. The alpha-CTD:promoter interactions seem to be absent in the RPo structure. Is the alpha-CTD resolved in the RPo structure at all?

6. Lines 99-109. Based on the comparison of the Thermus and Msm structures (pdb: 4XLN and 5TW1), the interaction mode between the two conserved arginine residues (R268 and R290 in Msm RNAP sigma A), the W-dayd residues, and the DNA nontemplate (nt) strand are largely consistent. The differences of interactions details described by the authors could reflect the arginine side chain rotamer adjustment in response to the subtle differences in relative positioning of the W-dayd and the nt strand to sigma A.

7. The authors proposed that the beta' insertion and the SigA-N together may block the entry of downstream DNA into the catalytic cleft during the transitioning from RPo to RPo. This is a very important hypothesis regarding the function of these two regions. Could the authors perform or propose some critical experiments that could possibly test this hypothesis?

Minor issues:

1. Define “Msm” (*M. smegmatis*) in the abstract.
2. Supplementary Figure 3 needs some position labeling for the nucleic acids.
3. The authors need to provide the PDB code for the 3.2-angstrom Msm RPo crystal structure.

Reviewer #3 (Remarks to the Author):

The mycobacteria RNA polymerase (RNAP) is an important antimicrobial drug target for against tuberculosis, a human disease claims almost 2 million lives annually. In this manuscript, Campbell, Darst and their colleagues reported structural analysis of mycobacteria RNAP with a full open promoter complex (RPo) at 3.2 Å-resolution. This new structure of a full Msm RPo contains RbpA/EσA and promoter DNA containing a complete transcription bubble and a 4-mer RNA hybridized to the DNA template strand in the RNAP active site. Their in-depth analyses leading to a comprehensive summary of the structural and functional features of mycobacteria RNAP that are similar to and distinct from Eco. In term of similarity, they observed a similar interaction mode between A/T rich upstream DNA and Msm RNAP α-subunit C-terminal domain (αCTD) or *E. coli* αCTD, suggesting a role of αCTD/UP-element interactions in mycobacteria transcription regulation. On the other hand, they found several unique features of mycobacteria transcription regulation. For example, they revealed the structure of a lineage-specific insert of the RNAP β' subunit that is unique to Actinobacteria. Furthermore, their structural analysis also reveals the disposition of the N-terminal segment of Msm σA, which may comprise an intrinsically disordered protein domain unique to mycobacteria. The clade-specific features of the mycobacteria RNAP provides clues to the profound instability of mycobacteria RPo compared with *E. coli*.

This work illustrates that while we have learned a lot from *Eco* transcription system regarding to structural and functional transcription paradigms, the *Eco* transcription system are not universally applicable among bacteria. This study also highlights the importance of studying phylogenetically distinct bacteria to gain a comprehensive insight into transcription and its regulation as well as to develop novel antibiotics to combat TB and other devastating bacterial diseases. This study would appeal to a broad readership of *Nature Comm*. I would recommend publication of the manuscript with following minor points corrected.

1. Page 6, line 112, Supp Figure 5, Personally, I feel this figure is really helpful for general readers to gain a comprehensive understanding of RNAP/promoter DNA interactions and orientates readers to understand the location of structural motifs in the rest of figures. If space is allowed, I would suggest to move this figure into main figure as Figure 1D.
2. Page 10, line 200, heading: “An Actinobacteria lineage-specific domain insertion in β' ”. It would be more clearer to spell out “β' subunit” instead.
3. Page 11, lines 226-227 “Group 1 σ's comprise three conserved structured

domains (σ_2 , σ_3 , and σ_4 ;) 41 but also an N-terminal extension (σ_{AN}) that in Eco contains σ_{70} 1.1 42. A defining characteristic of Group 1 σ 's is σ_{AN} , but the Group 1 σ 's vary greatly in length (residues N-terminal of conserved region 1.2: Mtb σ_{AN} , 225; Msm σ_{AN} 163; Eco σ_{70} N 95) and are not conserved across all clades (see below)."

Sentences seems a bit awkward. It can be rephrased as following:

"Group 1 σ 's comprise three conserved structured domains (σ_2 , σ_3 , and σ_4 ;) and one divergent N-terminal extension (σ_{AN}). Group 1 σ 's vary greatly in length (residues N-terminal of conserved region 1.2: Mtb σ_{AN} , 225; Msm σ_{AN} 163; Eco σ_{70} N, 95) and are not conserved across all clades (see below)".

Response to Reviewers' Comments:

Reviewer #1 (Remarks to the Author):

This manuscript reports a structure at 3.2Å of an *M. smegmatis* (Msm) initiation complex that extends from -37 to +13 with respect to the transcription start site and contains a promoter bubble and a short RNA “product”. The paper also contains further analysis of a previously published fork junction complex at 2.76Å. The 3.2Å structure is at higher resolution than any other initiation complex published previously, and it provides some useful new details about this complex. A major emphasis of the paper is an adventitious alpha CTD-DNA interaction in the fork-junction complex structure that mimics the natural alpha CTD interaction with UP element DNA in *E. coli* promoters. This interaction is between the alpha CTD in one RNAP with an A-rich DNA sequence downstream of the -35 hexamer bound to a neighboring symmetry-related RNAP. Although this interaction is not physiologically relevant, the actual details of the protein-DNA contacts are essentially identical to those in complexes with natural UP elements in *E. coli* transcription complexes, leading the authors to test whether there are natural UP elements in Msm. The authors explore the significance of UP element interactions by performing transcription and bioinformatic analyses that ultimately demonstrate the occurrence and functional importance of UP elements in mycobacterial promoters. The authors also report the structure of two lineage-specific parts of RNAP unique to Actinobacteria, an insert in beta prime and an N-terminal extension in the major sigma factor. I have only minor suggestions. There is plenty here to justify publication.

1. The Ebright group just published a structure of a *Mycobacterium tuberculosis* crystal structure of an open complex, Lin et al. 2017 Mol Cell. Although the emphases of the two papers are very different, two subjects that are discussed in both papers should be compared/contrasted. Specifically, it would be helpful to be more explicit about differences in the interpretations of the beta prime insertion as a gate and about the relevance of the N-terminal domain of sigma in Mtb to sigma 1.1.

Comparing our structural results with Lin et al (which was published while our manuscript was under review) – the structural results are essentially identical but the interpretation of the results is very different. We have added three paragraphs to the Discussion (lines 416–456) regarding this.

2. If I understand correctly, the Msm α CTD-DNA interaction described here is actually from the structure published previously, i.e. with the scaffold shown in Fig. S1 and Fig. 2, not with the scaffold shown in Fig. 1. The previous paper focused on the importance of a transcription factor and on the kinetic mechanism, but did not mention at all on the adventitious interaction between the promoter DNA just downstream of the -35 hexamer in one RNAP complex with the α CTD in the adjacent RNAP. The phrasing on page 6 is accurate, but it is sparse and on first reading it is unclear whether this is a new structure or the one published previously. It should be emphasized more clearly that this is not the physiologically relevant position for the α CTD-UP element interaction. On line 130, it should also be stated more clearly that D259 and E261 in α CTD and sigma R603 from *E. coli* RNAP are all conserved with the analogous residues in Msm RNAP and that if α CTD were bound to an UP element-like sequence upstream of the -35, these Msm residues would likely interact. However, in the adventitious interaction this cannot happen because sigma R603 is too far away. This situation doesn't really become clear until Fig. 5B.

The α CTD structure was built and refined as a part of the previous 2.76 Å us-fork structure (Hubin et al., 2017, *Elife* 6, e22520) but was not mentioned in the previous publication so as not to detract from the main focus of that manuscript, which was the structure and function of the activators CarD and RbpA. We have clarified the wording (lines 154-156).

The fact that the α CTD observed in the us-fork structure is bound to the DNA in a non-physiologically relevant position is clarified (lines 167-172), and the fact that *Msm* α CTD D253/D255 and σ^A R457 do not interact in our structure due to the non-physiological location of α CTD on the DNA is clarified (lines 176-180).

3. Line 21. Define Msm.

Msm is defined on line 18.

4. Line 140. The UP element bioinformatic analysis appears to have been done for just a proximal subsite. Was it also done with a distal subsite (i.e. a binding site for the 2nd α CTD)?

No – the position and conservation of the distal UP-element is not as clearly defined as the proximal UP-element (Estrem et al., 1998, *PNAS* 95, 9761). Given the relatively weak signal for the proximal UP-element we assumed it would be even more difficult to identify the distal UP-element using this approach.

5. Line 301. While many E. coli transcription experts may have worked under the assumption that the E. coli paradigm would apply to all bacteria, this sentence should be rephrased. It does a disservice to the many labs who long ago started studying transcription in other bacterial systems with just the opposite assumption, i.e. that there would be major differences. Given the 3 billion years of evolution since divergence of gram positives and gram negatives, for example, it is surprising how much similarity has survived!

This has been reworded (lines 393-394).

6. References 26 and 56 in the bibliography are incomplete.

These have been corrected.

7. Line 696. Since the sigma-aCTD contact does not actually occur in this complex, it should be made clearer that this interaction is inferred from the E. coli situation.

This has been clarified (lines 985-990).

8. Line 765: "analogous"

Corrected.

Reviewer #2 (Remarks to the Author):

The manuscript by Hubin et al. mainly focuses on the structural aspects of the mycobacteria transcription initiation complex (TIC). The primary content is based on a newly solved 3.2-angstrom crystal structure of an RNA polymerase (RNAP) open promoter complex (RPO) AND a previously reported 2.8-angstrom crystal structure of an RNAP-upstream fork DNA complex, both from Msm (*M. smegmatis*). These are the pioneer structures of mycobacteria RNAP and will benefit the understandings of related mycobacteria RNAPs and anti-mycobacteria studies. These two structures, solved at higher resolution than previously reported bacteria TIC structures, confirm the overall architecture and the protein:nucleic acid interaction framework of the complex, and provide better details. More importantly, this manuscript highlights a few previously unclarified features of the Msm RNAP: 1. The possible regulatory role of the alpha subunit CTD (a-CTD) through its interactions with the UP element; 2. The lineage-specific beta' subunit insertion (b'i1) and its possible contribution to mycobacteria transcription initiation; 3. The unique features of the Actinobacteria-specific sigma A N-terminal region (SigA-N) and its disposition in the context of TIC. While the a-CTD function is supported also from sequence conservation analysis and in vitro transcription data, the mechanisms underlying the b'i1 and SigA-N remain to be clarified.

Comments and concerns:

1. Because the Msm fork junction-structure (pdb: 5TW1) is of higher resolution and is described in detail, in particular for the interactions around the upstream edge of the TIC transcription bubble. The authors need to make it clearer that their previously published paper (ELife 2017) focuses more on the RbpA/CarD topic, while other important features of this structure is described for the first time in this manuscript.

We have added a clarifying statement (lines 87-88).

2. As the Msm fork-junction structure and the RPo structure may represent different critical stages of transcription initiation, it will be of great interest to describe the notable conformational differences between these two structures both in text and figure formats.

This statement was added (starting line 107): 'There were no significant conformational differences between the two structures, which superimposed with a root-mean-square deviation of 0.59 Å over 2,933 α -carbons.'

3. The nucleic acids conformation in the Msm RPo complex largely resembles that in Thermus RPo structure (pdb: 4XLN), and the primary difference is the disorder of the -11 to -7 region of the template (t) strand in the Msm structure. The interactions between this region and the RNAP may contribute to the maintenance of the open complex and are likely related to TIC bubble collapse as transcription progresses from initiation to elongation. Could the authors comment on possible reasons that may lead to the disorder of this t-strand region in the Msm structure, or any implications related to the disorder?

In the Msm RPo complex (3.2 Å resolution), the electron density for the t-strand DNA from -11 to -7 is too poor to model. In the Thermus RPo structure (pdb 4XLN, 4 Å resolution), there are actually two RPo complexes in the crystallographic asymmetric unit. In one of the non-crystallographically related complexes, the electron density for that region of the t-strand is poor but we were able to model the entire DNA (chain M). In the other complex, precisely the same region of the t-strand DNA (chain N), from -11 to -7, is disordered.

Eco RPo structures are also available from the Steitz group, one with σ^{70} (pdb 4YLN, 5.5 Å resolution) and one with σ^S (pdb 5IPL, 3.6 Å resolution). The very low resolution of the σ^{70} RPo makes comparisons with the other structures difficult – electron density for the t-strand DNA -11 to -7 is present but is poorly defined. In the σ^S -holo structure, the electron density for the t-strand DNA -11 to -7 is again broken and poorly defined.

So, in all of the RPo complexes available (*Eco*, Msm, Thermus), that region of the t-strand DNA appears to be mobile, giving rise to poor electron density.

Both Thermus and Msm RNAPs form relatively unstable Rpo compared to *Eco* RNAP (when tested on the same promoters, unpublished). For these reasons, we don't think

there are any conclusions we can draw from these observations relating t-strand order with RPo stability.

4. In the crystallography-related statistics (supplementary Table 1), R_{merge} and $I/\sigma I$ values (1.751 and 0.63) of the highest resolution shell do not meet general criteria. It is understandable that the authors want to push the resolution limit a bit for better observation of structural details. However, it will be then necessary to provide omit-type electron density maps to aid the appreciation of structural details, in particular in regions with novel information. 2Fo-Fc maps are provided in the supplementary materials mainly for the fork-junction structure but not for the newly reported RPo structure.

The general criteria for determining the high resolution cutoff for X-ray crystallography data are evolving. In brief, two papers (Diederichs & Karplus, 1997, *Nat Struct Biol* 4, 269; Karplus & Diederichs, 2012, *Science* 336, 1030) showed unequivocally the following:

I) The R_{merge} statistic used to evaluate data quality is 'seriously flawed'. 'Despite widespread use, it is poorly suited for determining the high-resolution limit and that current standard protocols discard much useful data'. Bottom line – R_{merge} is essentially useless in determining data quality, it should not be used – it is included in our tables because journals and reviewers normally insist that it be reported.

II) The standard criteria of $I/\sigma I > 2$ is also dramatically overconservative and results in the loss of useful crystallographic data. The reason this is important is NOT because one can then claim a higher resolution cutoff for ones structure, but because the additional higher-resolution data results in improved structural models (Karplus and Diederichs, 2012).

III) Karplus & Diederichs (Karplus and Diederichs, 2012) introduce an improved statistic, CC^* (essentially a Pearson correlation coefficient), that provides a single statistically valid guide for deciding which data are useful (i.e. data that improve the structural model).

Thus, the most important statistical quantity describing our data is not R_{merge} or $I/\sigma I$, but CC^* , which is 0.516 in the highest resolution shell (Supplementary Table 1) – this is a very high value that is significantly different from 0 (which would describe noise). Furthermore, the fact that our diffraction data contains significant information out to the highest resolution shell is given by the refinement statistics – note that R_{work} and R_{free} , and also C_{work} and CC_{free} (Karplus and Diederichs, 2012) do not diverge from one another in the highest resolution shell (as they would if the highest resolution shell contained noise).

Nevertheless, we have added Fig. 2A, which shows the electron density map for the nucleic acids in the 3.2 Å resolution RPo.

5. It is quite interesting that the interactions between the alpha-CTD and the promoter DNA region -29 to -24 are observed from a crystal packing-derived event. The authors performed sequence analysis and functional assays to reveal the possibility of transcription regulation by mycobacteria alpha-CTD through its interactions with the UP

element (upstream of -30 region), and generated a structural model with the alpha-CTD interacting with the UP element. These are great efforts toward the understanding of the mycobacteria alpha-CTD function. The alpha-CTD:promoter interactions seem to be absent in the RPo structure. Is the alpha-CTD resolved in the RPo structure at all?

The α CTD seems to be present in the same location in the RPo structure, but the electron density is extremely weak and broken (if we did not know to look there, we would not have noticed) and we did not attempt to model it. We added a clarifying statement (lines 154-156).

6. Lines 99-109. Based on the comparison of the *Thermus* and *Msm* structures (pdb: 4XLN and 5TW1), the interaction mode between the two conserved arginine residues (R268 and R290 in *Msm* RNAP sigma A), the W-dyad residues, and the DNA nontemplate (nt) strand are largely consistent. The differences of interactions details described by the authors could reflect the arginine side chain rotamer adjustment in response to the subtle differences in relative positioning of the W-dyad and the nt strand to sigma A.

As the reviewer points out, in the new *Msm* structures reported here, we observed interactions of two conserved Arg residues supporting the critical W-dyad interactions with the nucleic acids, but these Arg interactions were not observed in the previous *Thermus* RPo structures. We believe the likely explanation for this is the different crystallization conditions for the two complexes. The *Thermus* RPo was crystallized in a solution containing 1.6 M $(\text{NH}_4)\text{SO}_4$ (ionic strength ~ 4.9 M), while the *Msm* transcription initiation complexes were crystallized in a solution of polyethylene glycol and 0.2 M Li_2SO_4 (ionic strength ~ 0.6 M). The high ionic strength of the *Thermus* RPo crystallization solution likely weakened the Arg electrostatic interactions. This explanation for the differences in the *Thermus* and *Msm* structures was omitted due to space considerations, but has been added back (lines 139-146).

7. The authors proposed that the beta' insertion and the SigA-N together may block the entry of downstream DNA into the catalytic cleft during the transitioning from R_{Pc} to R_{Po}. This is a very important hypothesis regarding the function of these two regions. Could the authors perform or propose some critical experiments that could possibly test this hypothesis?

The reviewer is correct that these are important hypotheses to test but this would constitute an extensive set of experiments beyond the scope of the current manuscript.

Minor issues:

1. Define "Msm" (*M. smegmatis*) in the abstract.

Corrected

2. Supplementary Figure 3 needs some position labeling for the nucleic acids.

Added.

3. The authors need to provide the PDB code for the 3.2-angstrom Msm RPo crystal structure.

Added PDB IDs 5VI8 and 5VI5 (line 522-527).

Reviewer #3 (Remarks to the Author):

The mycobacteria RNA polymerase (RNAP) is an important antimicrobial drug target for against tuberculosis, a human disease claims almost 2 million lives annually. In this manuscript, Campbell, Darst and their colleagues reported structural analysis of mycobacteria RNAP with a full open promoter complex (RPo) at 3.2 Å-resolution. This new structure of a full Msm RPo contains RbpA/E σ A and promoter DNA containing a complete transcription bubble and a 4-mer RNA hybridized to the DNA template strand in the RNAP active site. Their in-depth analyses leading to a comprehensive summary of the structural and functional features of mycobacteria RNAP that are similar to and distinct from Eco. In term of similarity, they observed a similar interaction mode between A/T rich upstream DNA and Msm RNAP α -subunit C-terminal domain (α CTD) or E. coli α CTD, suggesting a role of α CTD/UP-element interactions in mycobacteria transcription regulation. On the other hand, they found several unique features of mycobacteria transcription regulation. For example, they revealed the structure of a lineage-specific insert of the RNAP β' subunit that is unique to Actinobacteria. Furthermore, their structural analysis also reveals the disposition of the N-terminal segment of Msm σ A, which may comprise an intrinsically disordered protein domain unique to mycobacteria. The clade-specific features of the mycobacteria RNAP provides clues to the profound instability of mycobacteria RPo compared with E. coli.

This work illustrates that while we have learned a lot from Eco transcription system regarding to structural and functional transcription paradigms, the Eco transcription system are not universally applicable among bacteria. This study also highlights the importance of studying phylogenetically distinct bacteria to gain a comprehensive insight into transcription and its regulation as well as to develop novel antibiotics to combat TB and other devastating bacterial diseases. This study would appeal to a broad readership of Nature Comm. I would recommend publication of the manuscript with following minor points corrected.

1. Page 6, line 112, Supp Figure 5, Personally, I feel this figure is really helpful for general readers to gain a comprehensive understanding of RNAP/promoter DNA interactions and orientates readers to understand the location of structural motifs in the rest of figures. If space is allowed, I would suggest to move this figure into main figure as Figure 1D.

We have added a new figure (Fig. 2), Part A shows the electron density map for the nucleic acids in the 3.2 Å-resolution RPo structure. Part B is the old Supplementary Fig. 5.

2. Page 10, line 200, heading: “An Actinobacteria lineage-specific domain insertion in β' ” . It would be more clearer to spell out “ β' subunit” instead.

Corrected

3. Page 11, lines 226-227 “Group 1 σ 's comprise three conserved structured domains (σ_2 , σ_3 , and σ_4 ;) 41 but also an N-terminal extension (σ_{AN}) that in Eco contains σ_{70} 1.1 42. A defining characteristic of Group 1 σ 's is σ_{AN} , but the Group 1 σ N's vary greatly in length (residues N-terminal of conserved region 1.2: Mtb σ_{AN} , 225; Msm σ_{AN} 163; Eco σ_{70} N 95) and are not conserved across all clades (see below).”

Sentences seems a bit awkward. It can be rephrased as following:

“Group 1 σ 's comprise three conserved structured domains (σ_2 , σ_3 , and σ_4 ;) and one divergent N-terminal extension (σ_{AN}). Group 1 σ N's vary greatly in length (residues N-terminal of conserved region 1.2: Mtb σ_{AN} , 225; Msm σ_{AN} 163; Eco σ_{70N} , 95) and are not conserved across all clades (see below)”.

Much better – corrected.